# Maize *GOLDEN2-LIKE* genes enhance biomass and grain yields in rice by improving photosynthesis and reducing photoinhibition

Xia Li[1,5], Peng Wang[2,3,5], Jing Li[1], Shaobo Wei[1], Yanyan Yan[1], Jun Yang[4], Ming Zhao[1], Jane A. Langdale [2] & Wenbin Zhou [1✉]

Photosynthetic efficiency is a major target for improvement of crop yield potential under agricultural field conditions. Inefficiencies can occur in many steps of the photosynthetic process, from chloroplast biogenesis to functioning of the light harvesting and carbon fixation reactions. Nuclear-encoded GOLDEN2-LIKE (GLK) transcription factors regulate some of the earliest steps by activating target genes encoding chloroplast-localized and photosynthesis-related proteins. Here we show that constitutive expression of maize *GLK* genes in rice leads to enhanced levels of chlorophylls and pigment-protein antenna complexes, and that these increases lead to improved light harvesting efficiency via photosystem II in field-grown plants. Increased levels of xanthophylls further buffer the negative effects of photoinhibition under high or fluctuating light conditions by facilitating greater dissipation of excess absorbed energy as heat. Significantly, the enhanced photosynthetic capacity of field-grown transgenic plants resulted in increased carbohydrate levels and a 30–40% increase in both vegetative biomass and grain yield.

[1] Institute of Crop Sciences, Chinese Academy of Agricultural Sciences, 100081 Beijing, China. [2] Department of Plant Sciences, University of Oxford, South Parks Road, Oxford OX1 3RB, UK. [3] CAS Center for Excellence in Molecular Plant Sciences, Institute of Plant Physiology and Ecology, Chinese Academy of Sciences, Shanghai, China. [4] Shanghai Key Laboratory of Plant Functional Genomics and Resources, Shanghai Chenshan Plant Science Research Center, Chinese Academy of Sciences, Shanghai Chenshan Botanical Garden, 201602 Shanghai, China. [5]These authors contributed equally: Xia Li, Peng Wang. ✉email: zhouwenbin@caas.cn

mproving photosynthetic efficiency has been proposed as a viable way to increase the yield potential of major crops, either through altering the type of photosynthetic pathway that is utilized[1,2] or by optimizing components of the existing pathways[3,4]. Crops use light as a source of energy for photosynthesis. However, crop leaves exposed to full sunlight absorb more light than they can use under agricultural field conditions. Under high light conditions, plants absorb excess light energy in the light-harvesting complex (LHC) that can lead to photooxidative damage and hence reduced photosynthetic efficiency, a process termed photoinhibition. Photoinhibition is an important limitation to photosynthesis under field conditions, and as such, physiological and molecular mechanisms underpinning the process have received extensive attention[5,6]. The main target of light stress is the photosystem II (PSII) complex that resides in the thylakoid membranes of chloroplasts. In optimal light conditions, the light captured by PSII is targeted to the photochemical reactions, including photosynthetic electron transport and $CO_2$ assimilation. In excess light, however, plants have to protect themselves from light damage either through dissipation of excess light energy captured by the light-harvesting antennae of PSII (LHCII) as heat, a process termed non-photochemical quenching (NPQ)[7], or through operation of a repair cycle that restores PSII structure, specifically restoration of degraded D1 protein[8]. Both non-photochemical and photochemical reactions are being targeted to improve photosynthetic efficiency in field-grown crops.

NPQ requires both conformational changes in the PsbS subunit of PSII and operation of the xanthophyll cycle to convert violaxanthin to zeaxanthin[9,10]. The xanthophyll cycle reversibly de-epoxidates violaxanthin to zeaxanthin via the intermediate antheraxanthin[11], with zeaxanthin being synthesized in high light. Zeaxanthin protects chloroplasts against photooxidative damage by acting as an antioxidant in the lipid phase of the thylakoid membrane[12]. Another xanthophyll, lutein, can directly quench excited chlorophyll ($^3$Chl*)[13]. A recent report showed that overexpression of PsbS protein in rice was sufficient to enhance NPQ in fluctuating light and that the modification led to increased biomass and grain yield in the field[14]. Similar experiments in tobacco demonstrated that photosynthetic efficiency and yield could be improved by altering both PsbS levels and operation of the xanthophyll cycle[15]. However, identical manipulations in Arabidopsis led to negative effects on growth and biomass accumulation even though photosynthetic efficiency and photoprotection were enhanced[16]. These findings suggest that manipulation of NPQ dynamics can enhance photosynthetic efficiency in different plant species but whether that enhancement converts into a yield increase is context dependent.

Successful attempts to modify photochemical reactions include the overexpression of Rieske FeS protein, which is a component of the cytochrome $b_6f$ (Cyt $b_6f$) complex that transfers electrons from PSII[17], and the manipulation of sedoheptulose-1,7-biphosphatase (SBPase) activity[18,19]. Increased levels of Rieske FeS protein resulted in increased electron transport rates, biomass, and seed yield in Arabidopsis[17], and engineered SBPase activity improved photosynthetic $CO_2$ assimilation, grain and biomass yield in wheat and tobacco[18,19]. These examples, plus those outlined for NPQ, demonstrate that the manipulation of individual steps in nonphotochemical or photochemical processes can improve photosynthetic efficiency, but the coordinated manipulation of multiple steps has not been well demonstrated in the field.

GOLDEN2-LIKE (GLK) transcription factors directly activate a large number of downstream target genes encoding chloroplastlocalized or photosynthesis-related proteins, including those required for chlorophyll (Chl) biosynthesis, light harvesting, and electron transport[20,21]. GLK proteins have been shown to regulate chloroplast development in all land plant species examined and to promote photosynthetic activity in previously non-green cells[20–26]. In rice, constitutive expression of maize GLK genes (ZmGLK1 or ZmG2) has been shown to induce chloroplast development in bundle sheath cells that are normally photosynthetically inactive; however, measurable rates of photosynthesis were not significantly different from wild type (WT) when plants were grown in greenhouse conditions[27]. Because GLK genes directly regulate the accumulation of PSII and electron transport components and also affect stomatal opening (at least in Arabidopsis)[20,28], we hypothesized that overexpression lines in rice might only outperform WT in conditions where efficient capture and subsequent direction of light energy into either photochemical or non-photochemical reactions was critical, i.e., under excessive light. If this were the case, differences in photosynthetic efficiency and yield would be revealed when plants were grown in fluctuating light conditions. To test this hypothesis, we grew transgenic rice lines that constitutively express either ZmGLK1 or ZmG2 in randomized plots in two field sites, over three growth seasons. Here we show that photoinhibition was reduced under high and fluctuating light conditions and that this decrease resulted in improved photosynthetic capacity, enhanced levels of carbohydrate accumulation, and increased biomass and grain yield.

## Results

**Increased photosynthetic capacity.** Transgenic rice lines designed to constitutively express maize GLK1 or G2 genes were generated by transforming constructs $ZmUBI_{pro}:ZmGLK1$ and $ZmUBI_{pro}:ZmG2$ into Oryza sativa spp. japonica cv. Kitaake. Two independent transgenic lines were isolated per construct. DNA gel blot analysis revealed that lines $ZmUBI_{pro}:ZmGLK1–2$, $ZmUBI_{pro}:ZmGLK1–3$, and $ZmUBI_{pro}:ZmG2–3$ had single copy insertions in the genome, whereas line $ZmUBIpro:ZmG2–2$ had two insertions (Supplementary Fig. 1a). The transgenes were expressed in all four lines, with the highest (and equivalent) transcript levels in $ZmUBI_{pro}:ZmGLK1–3$ and $ZmUBI_{pro}:ZmG2–3$ (Supplementary Fig. 1b).

To determine whether the downstream effects of GLK gene expression in rice are similar to those previously reported in Arabidopsis[20], blue native polyacrylamide gel electrophoresis (BN-PAGE) was first carried out. Consistent with the effects seen in Arabidopsis, levels of PSII supercomplexes, PSI/PSII dimers, and LHCII trimers were all increased in transgenic rice lines when compared to WT plants, particularly in line $ZmUBI_{pro}$:ZmG2–3 (Supplementary Fig. 2a). Immunoblotting with specific antibodies against representative subunits of photosynthetic thylakoid complexes further demonstrated increases in the abundance of PSI, PSII, Cyt $b_6f$, and LHCs of PSI and PSII but not in the levels of ATP synthase subunits and PsbS protein (Supplementary Fig. 2b). The observed differences in photosynthetic protein levels were mirrored by higher total Chl and carotenoid content in leaves (Fig. 1a) with levels of both pigments being significantly higher than WT in $ZmUBI_{pro}:ZmG2$ lines.

To evaluate whether photosynthetic capacity was enhanced in $ZmUBI_{pro}:ZmGLK1$ and/or $ZmUBI_{pro}:ZmG2$ rice lines, lightresponse and $CO_2$-response curves were generated for fieldgrown plants. Figure 1b shows that only small differences in net photosynthetic rate (Pn) were observed between WT and transgenic lines when photosynthetic photon flux density (PPFD) was <200 $\mu$mol m$^{-2}$ s$^{-1}$, whereas the differences were much more apparent at higher PPFD (Fig. 1b). Notably, Pn values for line $ZmUBI_{pro}:ZmG2–3$ were significantly higher than WT (as much as 48% higher at 1500 $\mu$mol m$^{-2}$ s$^{-1}$ light intensity). Concurrently, stomatal conductance and intercellular $CO_2$ concentration were significantly higher in transgenic lines than in WT plants, even at low light intensities (Fig. 1c, d). Apparent quantum yield (AQY)

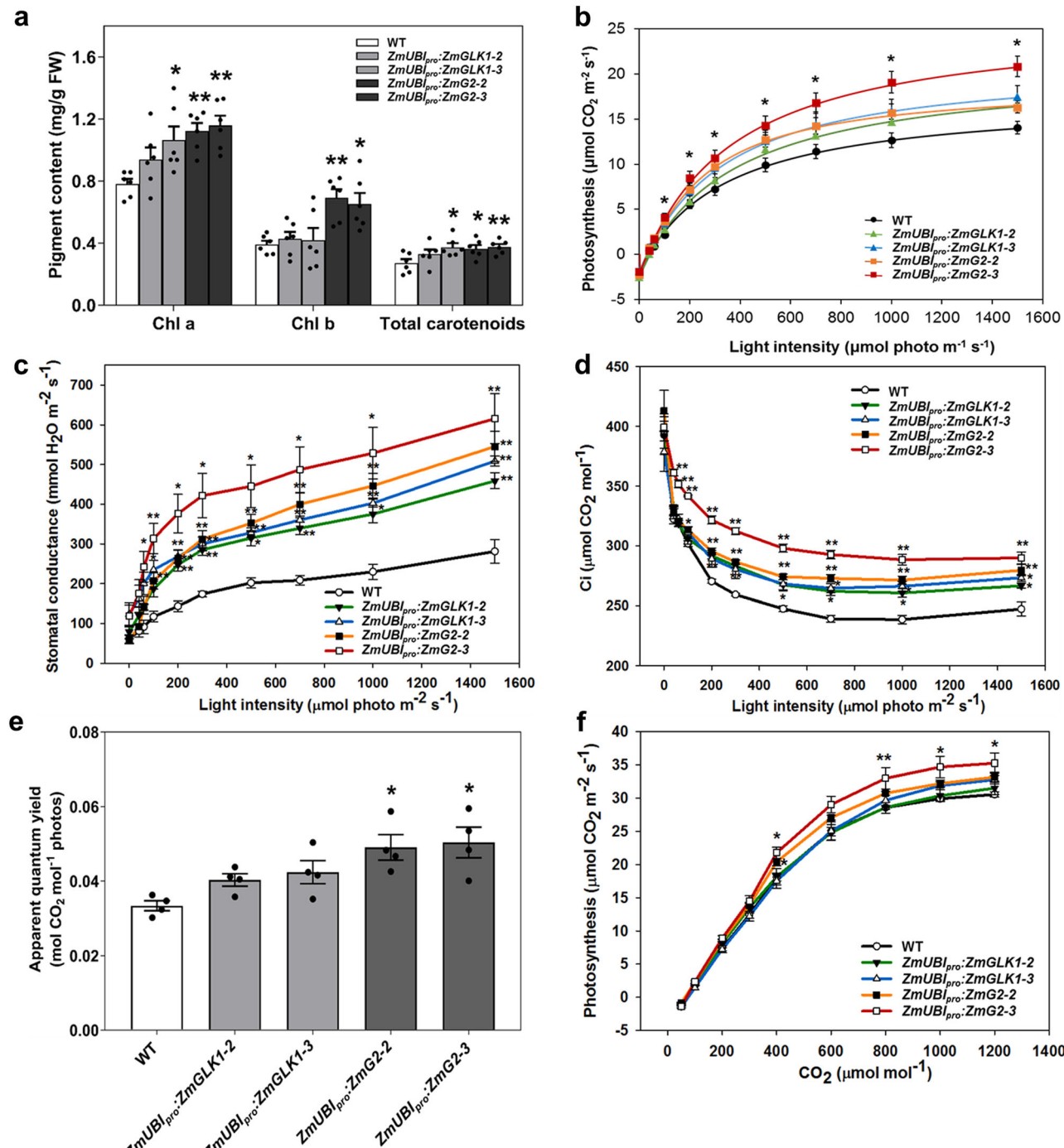

**Fig. 1 ZmUBI_pro:ZmGLK1 and ZmUBI_pro:ZmG2 transgenic lines exhibit higher rates of photosynthesis than wild-type plants when grown in the field.**
**a** Pigment content of flag leaves in WT and transgenic lines at the heading stage in the field in Beijing, 2018 ($n = 6$ biological replicates); **b–d** light response curve of net photosynthesis fitted by the FvCB model (**b**), stomatal conductance (**c**), and intercellular $CO_2$ concentration (Ci) (**d**) generated at 30 °C under normal air conditions in the field in Beijing, 2018 ($n = 4$ biological replicates). **e** Apparent quantum yield generated from fitted light response curves. Data are mean ± SE ($n = 4$ biological replicates). **f** $CO_2$ response curve of net photosynthesis generated at 1200 μmol m$^{-2}$ s$^{-1}$ PPFD and 30 °C in the field in Beijing, 2019. Data are mean ± SE ($n = 3$ biological replicates). Each dot represents a biological replicate. *$P < 0.05$, **$P < 0.01$ compared with WT according to a two-tailed Student's $t$ test.

values for $ZmUBI_{pro}:ZmG2$ transgenic lines were also higher than WT (Fig. 1e). Furthermore, significant higher value of Pn was seen in $ZmUBI_{pro}:ZmG2$ transgenic lines compared to WT in $CO_2$-response curves when the $CO_2$ concentration was >400 μmol mol$^{-1}$. The maximum carboxylation capacity ($V_{cmax}$), maximum electron transport rate ($J_{max}$), and triose phosphate utilization (TPU) rate

were all increased significantly in $ZmUBI_{pro}:ZmG2$ transgenic lines compared to WT, while the mitochondrial respiration (Rd) was mildly increased (Fig. 1f and Supplementary Table 1). Together these results demonstrate that constitutive expression of maize $GLK$ genes enhances photosynthetic capacity in rice, with $ZmG2$ having a greater effect than $ZmGLK1$.

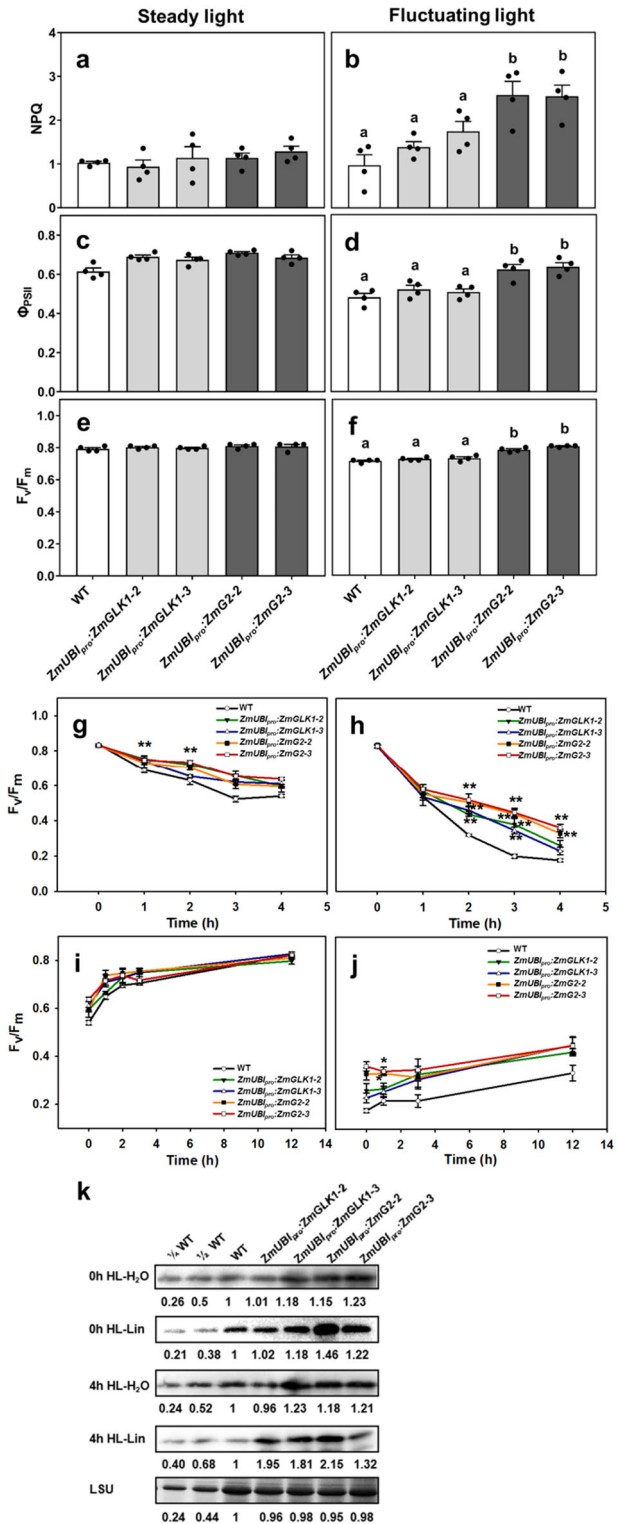

**Fig. 2 Elevated levels of D1 protein in $ZmUBI_{pro}$:$ZmGLK1$ and $ZmUBI_{pro}$: $ZmG2$ transgenic lines lead to better resistance to photoinhibition than wild-type plants in fluctuating light conditions. a–f** Non-photochemical quenching (NPQ) (**a, b**), quantum efficiency of photosystem II ($\Phi_{PSII}$) (**c, d**), and maximal PSII quantum efficiency ($F_v/F_m$) (**e, f**) under steady-state light (**a, c, e**) and after 3 days of treatment of fluctuating light (**b, d, f**) in WT and transgenic lines. Data are mean ± SE ($n = 4$ biological replicates), each dot represents a biological replicate. Different letters indicate a significant difference as determined by a one-way ANOVA test ($P < 0.05$). **g–j** Photoinhibition of PSII and recovery kinetics in WT and transgenic lines, including maximal PSII quantum efficiency ($F_v/F_m$) measured in detached leaves soaked in $H_2O$ under high light conditions (**g**); $F_v/F_m$ measured in detached leaves soaked in lincomycin under high light conditions (**h**); recovery of $F_v/F_m$ after photoinhibition in $H_2O$ (**i**) and 1 mM lincomycin (**j**). Data are mean ± SE ($n = 4$ biological replicates). *$P < 0.05$, **$P < 0.01$ compared with WT according to a two-tailed Student's $t$ test. **k** Immunoblot analysis of D1 protein in extracts from detached leaves of WT and transgenic lines before and after a 4-h exposure to high light (HL) in the presence (Lin) or absence ($H_2O$) of lincomycin. The Rubisco large subunit (LSU) was used as a loading control. The numbers below the gel lanes represent the relative protein level, which was quantified from the band intensity using the ImageJ software, and normalized relative to WT.

steady-state or fluctuating light conditions, and measured NPQ (which quantifies the ability to dissipate excess absorbed light energy as heat), $\Phi_{PSII}$ (which quantifies PSII photosynthetic efficiency), and $F_v/F_m$ (which quantifies maximal quantum yield of photosystem II). After 3 days of treatment, the response of WT and transgenic lines under steady-state light (200 μmol m$^{-2}$ s$^{-1}$) and fluctuating light were distinctly different. In steady-state light, NPQ, $\Phi_{PSII}$, and $F_v/F_m$ were similar in WT and transgenic lines (Fig. 2a, c, e). In fluctuating light, however, NPQ was increased dramatically in the transgenic lines compared to WT plants, whereas $\Phi_{PSII}$ was decreased from 0.61 to 0.48 in WT plants but dropped much less in $ZmUBI_{pro}$:$ZmG2$ transgenic lines (Fig. 2b, d). As was the case for pigment levels and light-response measurements (Fig. 1), NPQ, $\Phi_{PSII}$, and $F_v/F_m$ values for $ZmUBI_{pro}$: $ZmG2$ transgenic lines were significantly higher than WT after fluctuating light treatment (~150%, 30%, and 13% increase, respectively) (Fig. 2b, d, f). These results suggest that the transgenic lines can adapt to fluctuating light both by increasing heat dissipation and maintaining better PSII efficiency.

**Reduced photoinhibition under high light conditions.** Because transgenic lines displayed increased NPQ and higher $\Phi_{PSII}$ and $F_v/F_m$ than WT under fluctuating light conditions, we hypothesized that photoprotection may also be improved in high light conditions. To test this hypothesis, detached leaves were exposed to 4 h of high light stress (1200 μmol m$^{-2}$ s$^{-1}$), with or without 1 mM lincomycin (an inhibitor of chloroplast protein synthesis), and then allowed to recover at low light intensity (20 μmol m$^{-2}$ s$^{-1}$). Changes in $F_v/F_m$ were recorded as a measure of changes in maximal photosynthetic efficiency. Figure 2g shows that, in the absence of lincomycin, $F_v/F_m$ values in both WT and transgenic lines slowly decreased over the 4 h of high light exposure, with levels not dropping quite as low in transgenic lines (ending at 75% versus 87% of the dark-adapted values). The rates of recovery were also similar between WT and transgenic lines, with full recovery evident after 12 h (Fig. 2i). In the presence of lincomycin, however, the decline in $F_v/F_m$ in WT leaves was much more rapid, continuing until values approached 21% of the dark-adapted values after 4 h. In transgenic lines, especially $ZmUBI_{pro}$:$ZmG2$ lines, $F_v/F_m$ values were significantly higher than WT over the whole 4-h time period when lincomycin was present (Fig. 2h) and the recovery process was more efficient (Fig. 2j). Since

**Improved NPQ and $\Phi_{PSII}$ under fluctuating light conditions.** The enhanced photosynthetic rates observed in field-grown $ZmUBI_{pro}$:$ZmGLK1/G2$ lines (Fig. 1b) contrast with results reported for similar lines that were greenhouse grown[27] and reflect the observation that our transgenic $ZmUBI_{pro}$:$ZmG2$ plants grew much better than WT in field conditions but not in growth chambers. We thus hypothesized that the transgenic lines have an advantage over WT under natural light conditions, where light intensity fluctuates during the day. To test this hypothesis, we grew plants hydroponically in a growth chamber, either under

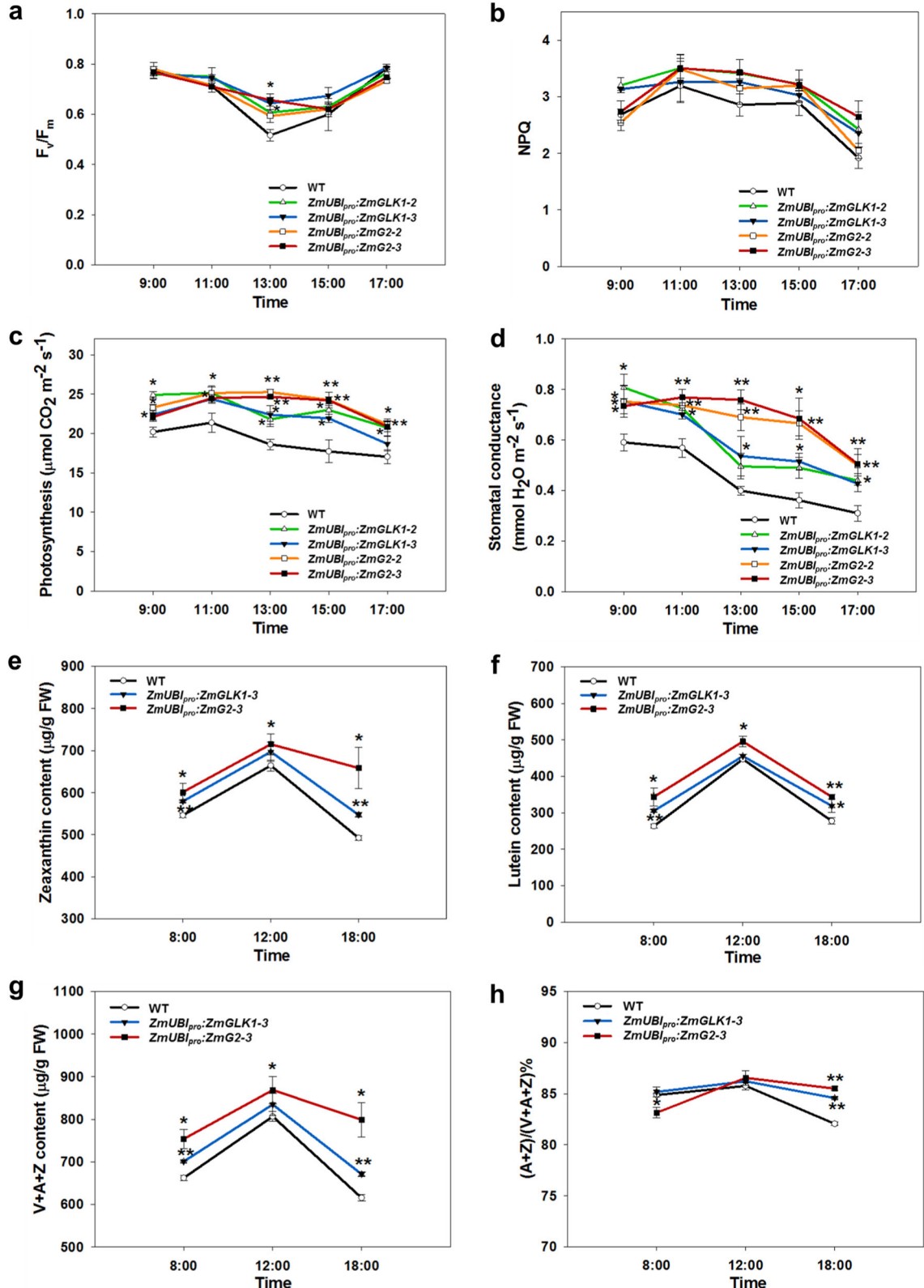

lincomycin blocks the repair of PSII by inhibiting de novo synthesis of D1 protein in the chloroplast, any decline in $F_v/F_m$ reflects the rate of photodamage to PSII. As such, the transgenic lines were more resistant to damage by high light than WT. Figure 2k shows that D1 protein levels were higher in the transgenic lines than WT, both before and after 4-h high light treatment, in both the presence and absence of lincomycin. Together these results suggest that the

resistance to photooxidative damage observed in transgenic lines expressing ZmGLK1 or ZmG2 is provided by elevated levels of PSII proteins.

To further evaluate the effects on photoinhibition in transgenic lines, we measured diurnal changes in photosynthetic parameters in the field. In the daytime, light intensity increases in the morning, rising to a maximum value at noon, and then decreases

**Fig. 3 Diurnal variation of photosynthetic parameters and xanthophyll pigments. a**, **b** Diurnal change in $F_v/F_m$ (**a**) and NPQ (**b**) values of flag leaves at the heading stage in the field in Beijing, 2018. Measurements were performed using a FluorPen. PPFD at each time point was 800, 1500, 2000, 900, and 350 μmol m$^{-2}$ s$^{-1}$, respectively. Data are mean ± SE ($n = 4$ biological replicates). **c**, **d** Diurnal curves of photosynthesis (**c**) and stomatal conductance (**d**) measured using a LICOR-6400 XT in the field from 9 a.m. to 5 p.m. in Hainan, 2019. PPFD at each time point was 600, 1200, 1500, 900, and 300 μmol m$^{-2}$ s$^{-1}$, respectively. All measurements were conducted with at least four biological replicates. Data are mean ± SE. **e**, **f** Diurnal change in zeaxanthin (**e**) and lutein (**f**) content. **g** Diurnal change in total content of xanthophyll pigments ($V + A + Z$). **h** Diurnal change in de-epoxidation state of the xanthophyll cycle calculated as the ratio $(A + Z)/(V + A + Z)$%. All pigments were measured in flag leaves sampled at the heading stage at 8 a.m., 12 a.m., and 6 p.m. from the field experiment in Beijing, 2019. Data are mean ± SE ($n = 3$ biological replicates). *$P < 0.05$, **$P < 0.01$ compared with WT according to a two-tailed Student's $t$ test.

to zero in the evening. The diurnal change curves of Chl fluorescence ($F_v/F_m$ and NPQ) were in accordance with changes in light intensity during the day, but both $F_v/F_m$ and NPQ were higher in transgenic lines than WT at the highest light levels (Fig. 3a, b). Photosynthetic rates of transgenic lines were significantly higher than WT throughout the day, and in the case of $ZmUBI_{pro}$:$ZmG2$ lines, rates did not decrease until late afternoon (Fig. 3c). Stomatal conductance was also maintained at higher levels in transgenic lines compared to WT, and in $ZmUBI_{pro}$:$ZmG2$ lines stomatal conductance did not even decline at midday when the sunlight was strongest (Fig. 3d).

**Increased zeaxanthin and lutein levels under high light.** Since carotenoids play essential roles in protecting plants from photodamage, we next investigated diurnal variation of carotenoid composition in the field. High-performance liquid chromatography (HPLC) analysis showed an increase in xanthophyll pigment levels (Fig. 3e, f, Supplementary Fig. 3) during the day in both $ZmUBI_{pro}$:$ZmGLK1$ and $ZmUBI_{pro}$:$ZmG2$ transgenic lines (particularly in $ZmUBI_{pro}$:$ZmG2$ lines), as well as in the pool size of the xanthophyll cycle (Fig. 3g). The de-epoxidation index of the xanthophyll cycle, calculated as the ratio $(A + Z)/(V + A + Z)$%, was induced quickly in $ZmUBI_{pro}$:$ZmG2$ lines before midday and after midday was higher than WT in both $ZmUBI_{pro}$:$ZmGLK1$ and $ZmUBI_{pro}$:$ZmG2$ transgenic lines (Fig. 3h). Zeaxanthin and lutein levels also increased significantly in $ZmUBI_{pro}$:$ZmG2$ transgenic lines before midday (Fig. 3e, f). Consistent with this observation, levels of β-carotene, which is upstream of zeaxanthin in the carotenoid biosynthesis pathway, decreased significantly between 8 a.m. and noon in both $ZmUBI_{pro}$:$ZmGLK1$ and $ZmUBI_{pro}$:$ZmG2$ transgenic lines (Supplementary Fig. 3d). Collectively, these results demonstrate that changes in NPQ in transgenic lines were correlated with changes in xanthophyll levels.

**Enhanced carbohydrate assimilation.** To determine how changes in photosynthesis influenced primary metabolism, we first examined starch accumulation in chloroplasts of WT and transgenic lines. Transmission electron microscopy revealed larger chloroplasts in bundle sheath cells of both $ZmUBI_{pro}$:$ZmGLK1$ and $ZmUBI_{pro}$:$ZmG2$ transgenic plants compared to WT (Supplementary Fig. 4a–f, i, j), as has previously been reported[27]. In addition, we noted that both the size and number of starch grains were significantly increased in both bundle sheath (Supplementary Fig. 4e, f) and mesophyll (Supplementary Fig. 4g, h) cell chloroplasts of $ZmUBI_{pro}$:$ZmG2$ transgenic lines as compared to WT, with a 16% increase in number and more than a 50% increase in size in mesophyll cells (Supplementary Fig. 4k, l).

To determine whether the elevated starch levels observed were generated as a consequence of increased starch formation during the day or reduced starch breakdown overnight, diurnal changes in total starch and sugar levels were measured in flag leaves of field-grown plants at three time points (7 a.m.–end of night, 12 a.

m.–midday, and 7 p.m.–end of day). Figure 4a shows that starch levels in leaves of $ZmUBI_{pro}$:$ZmG2$ lines were higher than WT throughout the day, with significant differences seen between noon and the end of day. The actual increases relative to WT were 20–33% at the end of the night, ~70% at noon, and 20–31% at the end of the day (Fig. 4a). Notably, levels of sucrose, glucose, and fructose were also higher than WT in $ZmUBI_{pro}$:$ZmG2$ lines, with significant differences observed at all time points (Figs. 4b–d). Metabolite profiling revealed elevated levels of other major metabolites, including amino acids and fatty acids (Fig. 4e, Supplementary Fig. 5a), but serine-to-glycine ratios were not altered (Supplementary Fig. 5b), suggesting that photorespiration is unaffected in transgenic plants. Collectively these data indicate that carbohydrate assimilation is enhanced in $ZmUBI_{pro}$:$ZmG2$ lines.

**Increased vegetative biomass and grain yield.** To evaluate the growth and yield potential of $ZmUBI_{pro}$:$ZmGLK1$ and $ZmUBI_{pro}$:$ZmG2$ rice lines in field conditions, we performed experiments in three consecutive growing seasons at two different locations (Beijing and Hainan) that have distinctive climates and day lengths (Supplementary Fig. 6). Beijing is warmer than Hainan throughout most of the season, and radiation levels are higher during the first half of the season. Figure 5 shows that, at both field sites, plant height was increased significantly in both $ZmUBI_{pro}$:$ZmGLK1$ and $ZmUBI_{pro}$:$ZmG2$ lines compared to WT (Fig. 5a, g), whereas tiller number was decreased (Fig. 5b, h). Flag leaf area (Fig. 5c, i) and length (Fig. 5d, j) were also consistently greater than WT, with differences being significant for $ZmUBI_{pro}$:$ZmG2$ lines at both sites. Flag leaf width was only significantly different from WT in $ZmUBI_{pro}$:$ZmG2$ lines grown at Hainan (Fig, 5e, k). Collectively, these differences translated into a significant increase in straw weight for $ZmUBI_{pro}$:$ZmG2$ plants at both locations (Fig. 5f, l).

For quantification of grain yield, a number of different measurements were made. In $ZmUBI_{pro}$:$ZmG2$ lines, panicle length, panicle weight, and seed number per panicle were all significantly increased over WT at both sites (Fig. 6a–c, f–h, k). Increases in all of these traits were also seen in $ZmUBI_{pro}$:$ZmGLK1$ lines, but only the increase in seeds per panicle was significantly different from WT at both sites (Fig. 6c, h). Despite the reduced tiller number, setting percentage, and 1000-seed weight (Supplementary Fig. 7a–d), a 19–27% increase in seeds per panicle in $ZmUBI_{pro}$:$ZmGLK1$ lines (Fig. 6c, h, k) translated into a 16–25% increase in seed yield per plant (Fig. 6d, i, l), and a 73–81% increase in $ZmUBI_{pro}$:$ZmG2$ transgenic lines (Fig. 6c, h, k) converted into a ~33% increase in seed yield per plant (Fig. 6d, i, l). Accordingly, the grain yield per plot increased by 13–15% and 28–32% in Beijing and by 14–18% and 34–45% in Hainan in $ZmUBI_{pro}$:$ZmGLK1$ and $ZmUBI_{pro}$:$ZmG2$ transgenic lines, respectively (Fig. 6e, j). The increased yields observed in Hainan were replicated with better performance (44–56% and 101–118% increase in $ZmUBI_{pro}$:$ZmGLK1$ and $ZmUBI_{pro}$:$ZmG2$ transgenic lines, respectively) in a subsequent field-growing season (Supplementary Table 2, Supplementary Fig. 7e). To exclude the

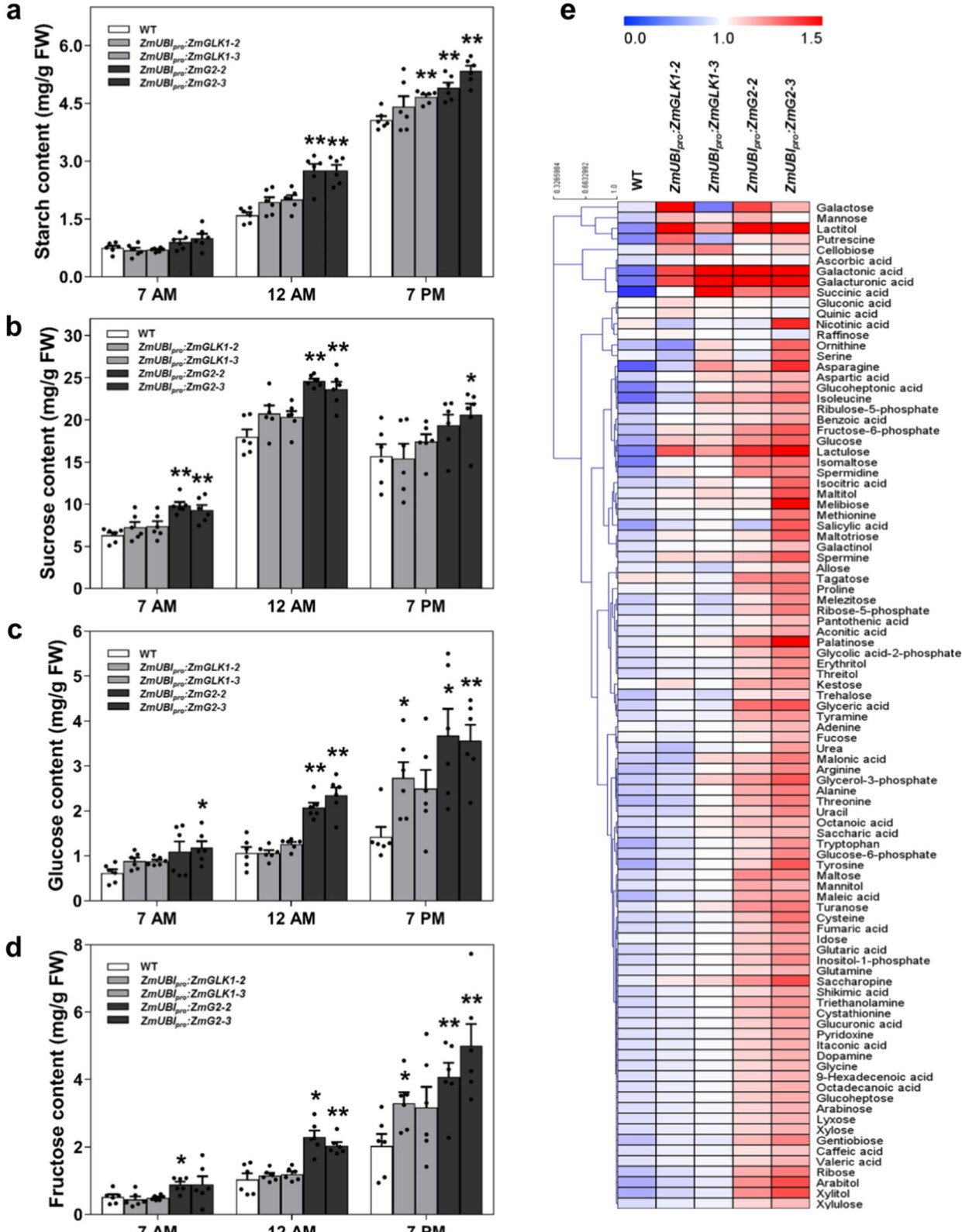

**Fig. 4 Leaves of *ZmUBI_pro:ZmG2* transgenic plants accumulate significantly higher levels of starch and sugars than wild-type plants. a–d** Starch (**a**), sucrose (**b**), glucose (**c**), and fructose (**d**) levels measured in flag leaves at the heading stage at 7 a.m., 12 a.m., and 7 p.m. in the field experiment in Hainan, 2019. Data are mean ± SE ($n = 6$ biological replicates), each dot represents a biological replicate. *$P < 0.05$, **$P < 0.01$ compared with WT according to two-tailed Student's $t$ test. **e** Hierarchical cluster analysis (HCA) of 95 primary metabolites in flag leaves at the heading stage in the field experiment in Beijing, 2018. Metabolite content is presented as median-centered averages with six biological replicates each. Red and blue colors indicate high and low content, respectively.

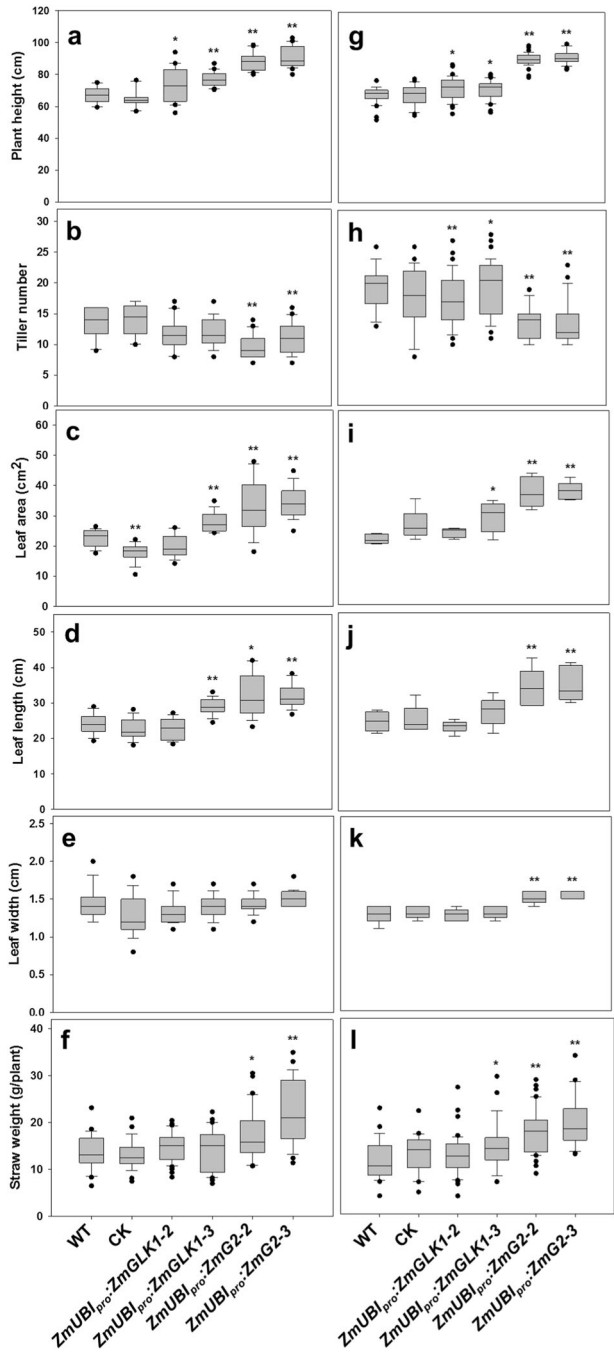

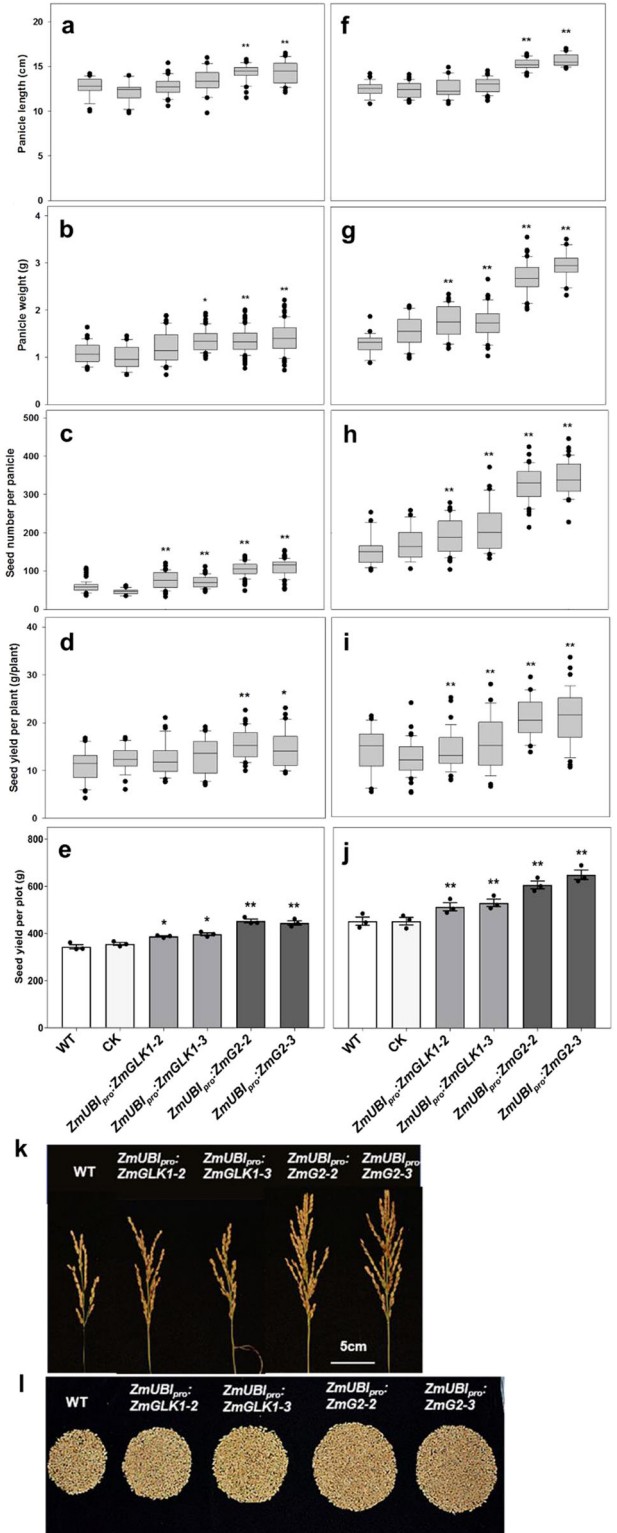

**Fig. 5 Increased vegetative biomass in *ZmUBI_pro*:*ZmGLK1* and *ZmUBI_pro*: *ZmG2* transgenic lines grown in Beijing and Hainan. a–f** Phenotypic parameters measured in the field experiment in Beijing, May 2018 to September 2018. All data were calculated from at least 20 independent rice plants. Data are mean ± SE. **g–l** Phenotypic parameters measured in the field experiment in Hainan, December 2017 to April 2018. **g**, **h**, **l** were calculated from at least 20 independent rice plants. **i**, **j**, **k** were calculated from at least five independent rice plants. CK = null segregants isolated from selfed heterozygous transgenic plants. Box and whisker plots show median (line) and outliers (black dots (●)). *$P < 0.05$, **$P < 0.01$ compared with WT according to a two-tailed Student's *t* test.

possibility that changes in hormone levels had caused the observed increases in plant biomass and seed yield, auxin, cytokinin, and gibberellin levels were quantified in the flag leaves

of field-grown plants. There were no statistical differences observed between WT and transgenic lines for any of the three (Supplementary Fig. 8a–c). Taken together, these data demonstrate that enhanced photosynthetic capacity led to increased biomass production and grain yield for *ZmUBI_pro*:*ZmGLK1* and *ZmUBI_pro*:*ZmG2* transgenic lines at both locations, with a much better performance in Hainan.

**Fig. 6 Enhanced grain yield in *ZmUBI_pro*:*ZmGLK1* and *ZmUBI_pro*:*ZmG2* transgenic lines in Beijing and Hainan. a–e** Yield parameters obtained from the field experiment in Beijing, May 2018 to September 2018. **f–j** Yield parameters obtained from the field experiment in Hainan, December 2017 to April 2018. All data except **e** and **j** were calculated from at least 20 independent rice plants. Seed yield per plot (**e**, **j**) was calculated from 30 independent rice plants within a plot and three plots that were placed randomly in the field. CK = null segregants isolated from selfed heterozygous transgenic plants. Box and whisker plots show median (line) and outliers (black dots (●)). In **e**, **j**, data are mean ± SE ($n = 3$ replicates). *$P < 0.05$, **$P < 0.01$ compared with WT according to a two-tailed Student's $t$ test. **k**, **l** Comparison of single panicle (**k**) and seed yield per plant (**l**) of WT and transgenic plants from the field experiment in Hainan, December 2017 to April 2018. Scale bars = 5 cm.

## Discussion

We have shown that constitutive expression of *ZmGLK1* or *ZmG2* in rice leads to elevated levels of Chl, carotenoid, and xanthophyll cycle pigments and to increased levels of some PSII components (Figs. 1 and 3). These alterations were associated with reduced photoinhibition when plants were grown in high or fluctuating light conditions (Fig. 2). Transgenic lines also showed higher photosynthetic efficiency, with increased stomatal conductance and intercellular $CO_2$ concentrations (Figs. 1 and 3), even under strong light intensities at midday in the field (Fig. 3h). Enhanced photosynthetic efficiency together with reduced photoinhibition led to significantly increased carbohydrate accumulation and improved biomass and grain yields when the *ZmGLK1* and *ZmG2* overexpression lines of rice were grown in the field (Figs. 4–6, Supplementary Fig. 5).

The photosynthetic phenotype exhibited by *ZmGLK* overexpression lines, particularly *ZmUBI_pro*:*ZmG2* lines, suggests that enhanced *GLK* gene function provided a multi-layered buffering effect against photooxidative damage that can be caused by high or fluctuating light. The first buffer was provided by elevated NPQ. Although transgenic rice plants overexpressing PsbS protein have previously been shown to display higher NPQ and reduced photoinhibition in the field (and thus enhanced grain yield and biomass)[14], PsbS protein levels were not altered in the lines examined here (Supplementary Fig. 2b). However, field-grown *ZmGLK* overexpression lines accumulated increased xanthophyll and lutein throughout the day and had a higher de-epoxidation ratio of xanthophyll cycle components after the noontime high light exposure (Fig. 3, Supplementary Fig. 3). As such, we conclude that increased levels of xanthophylls, including zeaxanthin and lutein, likely enhanced resistance to photoinhibition by facilitating more heat dissipation of extra absorbed light[7,29–31]. The second buffer was provided by PSII being more resistant to damage. If photooxidative damage occurs in PSII, the PSII repair cycle operates to restore levels of the D1 polypeptide and maintain photosynthetic activity[32,33]. In *ZmGLK* overexpression lines, the higher levels of LHC and D1 proteins may have kept the complex functional for longer than in WT and allowed faster recovery after high light or fluctuating light-induced damage. Given that it has previously been shown that the maximal efficiency of PSII is positively correlated with the abundance of functional PSII centers[34,35], the hyperaccumulation of PSII pigment–protein antenna complexes in *ZmGLK1* and *ZmG2* overexpression lines would both enhance the efficiency with which light was captured and reduce photoinhibition.

Consistent with previous reports that showed *GLK* genes promote stomatal opening and thus enhance stomatal conductance in *Arabidopsis*[28], we showed here that *ZmGLK*

overexpression in rice resulted in increased stomatal conductance and intercellular $CO_2$ concentrations in plants grown in the field (Figs. 1 and 3). These increases were associated with improved photosynthetic rates (Figs. 1 and 3) and significantly increased accumulation of carbohydrates (both starch and sugars) in *ZmUBI_pro*:*ZmG2* lines (Fig. 4, Supplementary Fig. 5). Given that sugars are transported symplastically in rice[36], yield increases can only be realized if any increase in $CO_2$ assimilation is accompanied by enhanced loading of photosynthate into the phloem, for transport to sink leaves and ultimately for grain filling. Indeed, plasmodesmatal conductance has been shown to play an important role in regulating sugar translocation and grain filling in rice[37]. Despite GLK function not having any known direct effect on plasmosdesmata, it was previously shown that constitutive expression of *ZmGLK* genes in rice leads to higher numbers of plasmodesmatal connections between bundle sheath and mestome cells and between bundle sheath and mesophyll cells[27]. The combined effect of GLK activity on stomatal and plasmodesmatal conductance may thus lead to enhanced photosynthetic rates and yield in *ZmUBI_pro*:*ZmG2* lines through more efficient photosynthesis in mesophyll cells accompanied by more effective phloem loading of sugars, by additional photosynthetic capacity in bundle sheath cells, or by a combination of both.

The changes in photosynthesis observed in *ZmUBI_pro*:*ZmG2* lines led to grain yield increases of at least 30–40% (Fig. 6), with increases relative to WT higher in the Hainan field than in Beijing. This is most likely because radiation levels and temperature were both higher in Hainan during grain filling (Supplementary Fig. 6), the stage at which a positive correlation between yield and solar radiation level has been established[38,39]. In traditional breeding, the generation of hybrid rice lines led to yield increases up to 50% per unit area compared with inbred lines[40,41]. Genetic manipulation has also previously generated notable increases in grain yield[42,43]. For example, overexpressing the HYR gene in rice yielded 29% more grain, with increased photosynthesis and resistance to heat stress[44]; introduction of a new chloroplastic photorespiratory bypass increased rice yield by 7–27%[45]; overexpression of the MADS-box transcription factor *zmm28* resulted in maize plants with increased plant growth, photosynthesis capacity, and nitrogen utilization in the field[46]; and similar effects were reported for rice lines overexpressing GRF4 transcription factor[47]. At face value, the yield increases reported here are thus similar to, if not better than, those reported previously. However, direct comparisons are difficult because of the different genetic backgrounds and different field locations used in each case. Notable in this regard is the opposite impact on yield reported in tobacco (higher) and *Arabidopsis* (lower) when PsbS and the xanthophyll cycle were manipulated to alter NPQ kinetics[15,16]. The inconsistent effect between species may be attributed to interspecies differences in source–sink capacity and/or different trade-off strategies associated with exogenous gene expression. In this study, overexpression of maize GLK transcription factors in the model rice cultivar Kitaake improved multiple steps in the photosynthetic process and led to a significant increase in grain yield. Future work will need to investigate the extent to which constitutive GLK activity can increase yield in an elite rice germplasm and determine whether the effect of GLK activity on crop yield can be applied more generally to other species.

## Methods

**Gene cloning and construct design**. Full-length complementary DNAs (cDNAs) of *ZmG2* (GenBank accession number AF318579) and *ZmGLK1* (GenBank accession number AF318580) were amplified by PCR from cDNA clones isolated previously[48]. Five μl 2×Phusion® High-Fidelity PCR Master Mix (Thermo Fisher Scientific) and 3.5 μl 4 M Betaine were used per 10 μl PCR reaction. PCR conditions were: 98 °C for 3 min; 35 cycles of 98 °C for 15 s, 63 °C for 40 s, 72 °C for 1 min; and 72 °C for 10 min. The coding sequences were subcloned into Gateway® donor

vector pDONR™207 through a BP reaction, sequenced, and then cloned downstream of the *ZmUBI* promoter in the binary destination vectors pSC310 (kindly gifted by Julian Hibberd, University of Cambridge, UK) or pVec8-Gateway[49], via LR reactions. Four constructs were produced: pFPW57CC (pVec8-*ZmG2*), pFPW58CC (pVec8-*ZmGLK1*), pSC310–57E, and pSC310–58E. The PCR primer pairs are given in Supplementary Table 3.

**Kitaake rice transformation.** *O. sativa* spp. *japonica* cultivar Kitaake calli induced from mature seeds were used for transformation with *Agrobacterium tumefaciens* strain EHA105 carrying the construct of interest. Callus induction, transformant selection, and seedling regeneration were performed at 32 °C under continuous light according to a protocol (available from https://langdalelab.files.wordpress.com/2018/06/kitaake-rice-trans-formation.pdf) modified from ref. [50]. Hygromycin-resistant regenerants were screened by PCR for *ZmG2* or *ZmGLK1*, and positive T0 seedlings were transplanted into soil. Plants were grown in a transgenic greenhouse with a day/night temperature of $30/22 \pm 3$ °C and a diurnal light regime of 16-h light (supplemented to ~300 µmol m$^{-2}$ s$^{-1}$) and 8-h dark, and T1 seeds were collected. The PCR primer pairs are given in Supplementary Table 3.

**Rice growth conditions.** For hydroponic culture, rice seedlings were grown as previously reported[51] in modified Kimura B solution, which contained 0.5 mM $(NH_4)_2SO_4$, 0.54 mM $MgSO_4 \cdot 7H_2O$, 1 mM $KNO_3$, 0.3 mM $CaCl_2$, 0.18 mM $KH_2PO_4$, 0.09 mM $K_2SO_4$, 16 µM $Na_2SiO_3 \cdot 9H_2O$, 9.14 µM $MnCl_2 \cdot 4H_2O$, 46.2 µM $Na_2MoO_4 \cdot 2H_2O$, 0.76 µM $ZnSO_4 \cdot 7H_2O$, 0.32 µM $CuSO_4 \cdot 5H_2O$, and 40 µM Fe(II)-EDTA, with the pH adjusted to 5.8. The nutrient solution for culture was renewed every 3 days. The temperature of the growth chamber was maintained at 28 °C and humidity at ~70%, the photoperiod was 14-h light/10-h dark, ~200 µmol m$^{-2}$ s$^{-1}$ photon intensity.

Field experiments were conducted in paddy fields at two experimental stations, between 2016 and 2019, using homozygous seeds of at least the T4 generation. For natural long-day conditions, plants were grown in Beijing (northern China, temperate climate, N40°13′49.86″, E116°33′28.23″, day length >15 h) from May to September of both 2017 and 2018 growing seasons. For natural short-day conditions, plants were grown in Hainan, the South China Experimental Station of the Institute of Crop Sciences of CAAS (southern China, Sanya, Hainan Province, tropical climate, N18°23′25.61″, E109°11′42.22″, day length <12 h) from December 2016 to April 2017, December 2017 to April 2018, and December 2018 to April 2019. Fertilizer application, pesticide use, and all other field management practices were carried out in the same way at both sites. Rice seedlings were transplanted in plots of 20 rows ×10 plants, with spacing of 20 cm between rows and between plants. The border rows were excluded in sampling and measurements. Three plots were planted as biological replicates, and all lines were completely randomized in each plot. For the final field test, the edge lines of each plot were removed to avoid margin effects.

**Photosynthetic measurements in the field.** Gas exchange measurements were carried out using a LI-COR 6400XT portable photosynthesis system (LI-COR Biosciences, Lincoln, USA). Flag leaves of plants at the filling stage were used for the determination of light-response and $CO_2$-response curves in Beijing. Leaves were acclimated in the chamber for approximately 30 min before measurements were made on the mid portion of the leaf blade. For the light-response curve, $CO_2$ concentration was set at 400 µmol mol$^{-1}$, and the PPFD was set from 1500 to 0 µmol m$^{-2}$ s$^{-1}$. Light-response curves were fitted using the Farquhar–von Caemmerer–Berry model, and AQY was calculated as the initial slope of the light response curve at light intensity <200 µmol m$^{-2}$ s$^{-1}$ [52]. For the $CO_2$-response curve, the PPFD was set at 1200 µmol m$^{-2}$ s$^{-1}$, and the $CO_2$ concentration was gradually decreased from 400 to 50 µmol mol$^{-1}$ and then increased from 400 to 1200 µmol mol$^{-1}$. The maximum rates of Rubisco-mediated carboxylation ($V_{cmax}$), electron transport ($J_{max}$), and Rd were calculated from the $CO_2$-response curve according to previously published methods, considering the improved temperature response, with the $O_2$ concentration set as 210 µmol mol$^{-1}$ and leaf absorbance as 0.93[53,54]. The $V_{cmax}$ and $J_{max}$ reflect Rubisco activity and the maximum rate of electron transport used in the regeneration of ribulose-1,5-bisphosphate, respectively. Rate of TPU was calculated from A-Ci curves as previously reported[55]. Diurnal changes in photosynthesis and leaf Chl fluorescence ($F_v/F_m$ and NPQ) were measured from 9 a.m. to 5 p.m. in the field using a LI-COR 6400XT or FluorPen FP100 (Photon Systems Instruments, Czech Republic). Leaves were dark adapted for 15–20 min prior to $F_v/F_m$ and NPQ measurements. Chl and carotenoid contents were measured spectrophotometrically according to published protocols[56].

**Fluctuating light treatment.** Three-week-old rice seedlings grown hydroponically under normal light (200 µmol m$^{-2}$ s$^{-1}$) were exposed to computer-controlled fluctuating light conditions of 1200 µmol m$^{-2}$ s$^{-1}$ for 1 min and 100 µmol m$^{-2}$ s$^{-1}$ for 4 min. After 3 days of exposure, the youngest fully expanded leaves were used to measure $F_v/F_m$, NPQ, and $\Phi_{PSII}$ using a Dual-PAM 100 (Walz, Germany). Plants were kept in the dark for 20 min before all Chl fluorescence measurements.

**Photoinhibition and recovery treatments.** The sensitivity of PSII to high light stress was measured as the change of $F_v/F_m$ over exposure time, using the youngest leaf of 5-week-old rice seedlings grown in hydroponics. The detached leaves were adapted by soaking in ddH$_2$O or 1 mM lincomycin at room temperature for 3 h (light intensity 20–30 µmol m$^{-2}$ s$^{-1}$) and were then exposed to 4-h light intensity at 1200 µmol m$^{-2}$ s$^{-1}$ (LED lights). $F_v/F_m$ was measured every hour with a FluorPen FP100 (PSI, Czech) after dark adaptation for 15–20 min. After 4 h, recovery from photoinhibition was assessed by transferring leaves to low light conditions (20 µmol m$^{-2}$ s$^{-1}$) for 12 h. Leaf samples were frozen in liquid nitrogen immediately after $F_v/F_m$ was measured. Total proteins were extracted and separated by sodium dodecyl sulfate-polyacrylamide gel electrophoresis (SDS-PAGE), and then subjected to immunoblotting using D1 antibody (Agrisera). The relative protein level was quantified from the band intensity using the ImageJ software and normalized relative to the WT.

**Transmission electron microscopy.** Rice leaves were cut into small pieces and fixed under vacuum in 0.1 M phosphate buffer (pH 7.2), 2% (v/v) glutaraldehyde, 0.5% (w/v) paraformaldehyde, and then further fixed for 4 h in phosphate-buffered saline containing 2% $OsO_4$ (pH 7.2). After dehydration in an ethanol series, samples were embedded in LR White resin (London Resin, Berkshire) and then sectioned using an ultramicrotome (Power TomeXL; RMC, AZ). Ultrathin sections (50–70 nm) were double stained with 2% (w/v) uranyl acetate and 2.5% (w/v) lead citrate aqueous solution before observation. Sections were examined with a Hitachi H-7650 transmission electron microscope. Numbers and size of chloroplasts and starch grains were quantified by ImageJ as previously described[57].

**Sugar and starch content measurement.** Leaf starch and soluble sugars were isolated from frozen leaves. Approximately 50 mg ground material was extracted twice with 1 ml 80% (v/v) ethanol at 80 °C for 30 min. Supernatants were combined and dried overnight in a speedvac (Eppendorf) at room temperature and then dissolved in deionized water. Sugars were measured using the Sucrose, D-fructose, and Glucose Assay Kit according to the manufacturer's (Megazyme) protocol. The rest sediments were used for starch measurements using the Total Starch Assay Kit, again following the manufacturer's (Megazyme) protocol.

**Analysis of agronomic traits.** Important agronomic traits, including plant height, leaf area, leaf length, leaf width, straw weight, tiller number, single panicle weight and length, seed number per panicle, seed-setting rate, and grain yield per plant, were measured on a single-plant basis. Plant height was determined as the height of the main tiller. Leaf area, leaf length, and leaf width were measured using a LI-3000C portable leaf area meter (LI-COR Biosciences, Lincoln, USA). Single panicle weight and length plus seed number per panicle were all measured for the main panicle. Filled and unfilled grains of the main panicle were separated manually to calculate seed-setting rate (filled grains/(filled + unfilled grains) × 100%). All filled grains from a single plant were collected and dried at 50 °C in an oven for measurements of grain yield per plant. Randomly picked filled grains were used for 1000-grain weight measurements. All grains in a single plot were collected and treated as described above for measurements of actual yield per plot.

**DNA gel blot analysis.** Gel blot analysis was performed as previously reported[27]. Total plant DNA was extracted from fresh leaf tissue by a cetyltrimethylammonium-bromide-based method[58]. For each transgenic line, 10–15 µg genomic DNA was digested with BglII restriction endonuclease (NEB) at 37 °C overnight. Digested DNA was electrophoresed on a 1% agarose gel for 6 h at 50 V and then transferred to positively charged nylon membrane (Roche). Blots were hybridized with digoxigenin (DIG)-labeled probes synthesized with hygromycin primers (Supplementary Table 3) and the PCR DIG Probe Synthesis Kit (Roche Diagnostics). Hybridization signals were detected using the CDP-Star reagent (Roche). Pre-hybridization, hybridization, washing, and detection were performed according to Roche's DIG Application Manual.

**Quantitative reverse transcriptase PCR (RT-PCR).** The youngest fully expanded leaves of 3-week-old hydroponically grown rice seedlings were harvested and immediately frozen in liquid nitrogen. Samples were ground and RNA was extracted with TRIZOL reagent (Invitrogen). The quality and quantity of RNA was assessed with the NanoDrop-1000 (NanoDrop, USA). After DNase treatment (TURBO DNA-free Kit, Ambion, USA), 1 µg total RNA was used as template to synthesize first-strand cDNA with SuperScript® III Reverse Transcriptase (Invitrogen). PowerUp SYBR Green Master Mix (Applied Biosystems) was used for quantitative RT-PCR with the ABI QuantStudio 6 Flex instrument (Applied Biosystems, USA). The $2^{-\Delta\Delta CT}$ method was used to determine the relative transcript levels[59]. Transcript levels for each gene were normalized to the levels of *OsActin*. PCR primer pairs are given in Supplementary Table 3.

**Blue native PAGE.** Thylakoid membranes were separated as previously described[15,60] with the following modifications. In all, 0.5–1 g frozen rice leaves from 4-week-old rice seedlings grown hydroponically were homogenized in ice-cold isolation buffer (400 mM sucrose, 10 mM NaCl, 2 mM $MgCl_2$, 50 mM HEPES/NaOH, pH 7.8) and then filtered through two layers of Miracloth (Millpore, USA). After centrifuging at 5000 × *g* for 10 min at 4 °C, the pellet was re-suspended in ice-cold wash buffer (330 mM sorbitol, 50 mM Bis-Tris-HCl, pH 7.0) twice and centrifuged

at $5000 \times g$ for 5 min at 4 °C. The pellet was finally re-suspended in ice-cold suspension buffer (25 mM Bis-Tris-HCl, 20% glycerol, pH 7.0). The Chl concentration of thylakoid membranes was measured following extraction with 80% acetone[61]. Isolated thylakoid membranes (10 µg Chl at 0.5 mg Chl ml$^{-1}$) were solubilized with 2% dodecyl-β-D-maltoside (Sigma) for 30 min on ice with gentle agitation at intervals. The solubilized membranes were centrifuged at $14,000 \times g$ for 15 min at 4 °C. The supernatant proteins were mixed with 1/10 volume of Coomassie sample buffer (5% Coomassie blue G-250, 100 mM Bis-Tris, 500 mM 6-aminocaproic acid, and 30% glycerol (v/v)). BN-PAGE was performed using 4–13% Bis-Tris gels to separate photosynthetic complexes according to previously described methods[62].

**Immunoblot analysis**. Frozen leaves from 4-week rice seedlings grown in hydroponics were ground to a fine powder and homogenized with 200 µl extraction buffer containing 20 mM Tris (pH 7.5), 100 mM NaCl, 2.5 mM MgCl₂, 1 mM EGTA, 1 mM dithiothreitol, and protease inhibitor cocktail (1:50) (Roche). The homogenate was gently shaken on ice for 30 min and then centrifuged at $12,000 \times g$ for 20 min at 4 °C. The supernatants were used for the following steps. After quantifying protein content using the Quick Start™ Bradford Reagent (Bio-rad), protein was mixed with 6× protein loading buffer (Tiangen) and boiled for 5 min. Fifteen µg protein was loaded per lane on 3–12% SDS-PAGE gels and then electrophoresed for 30 min at 80 V followed by 120 V for 1 h. Proteins were transferred to Amersham™ Hybond™ PVDF membrane (GE Healthcare) in transfer buffer (25 mM Tris, 192 mM glycin, pH 8.3). After transfer, membranes were blocked with 5% skim milk in TBST buffer (20 mM Tris/HCl, pH 7.6, 137 mM NaCl, 0. 1% Tween) overnight at 4 °C. After washing, membranes were incubated for 1 h at room temperature with primary antibodies at different dilutions as recommended by the manufacturer (Agrisera) and then incubated with secondary antibody (goat anti-rabbit) at a dilution of 1:20,000 for 1 h at room temperature. Membranes were then washed and incubated for 5 min at room temperature in ECL solution (Amersham, GE Healthcare). Membranes were imaged using a Biostep Celvin S Chemiluminescence Imager device (Biostep).

**Pigment content measurement**. Flag leaf samples were taken at heading stage and immediately frozen in liquid nitrogen. Ground leaf samples were extracted in 100% acetone for Chl and carotenoid content measurement by spectrophotometric analysis. In addition, 0.1% BHT–ethanol was used for extraction of xanthophyll pigments. Extracts were filtered through a 0.22 µm membrane filter, separated on a C18 column (Agilent), and then 10 µl aliquots were quantified by HPLC (Agilent 1200, USA) with a VWD/DAD detector at 445 nm wavelength. The pigments were eluted at a flow rate of 1 ml min$^{-1}$ at a column temperature of 25 °C using solvent A (acetonitrile: methanol: water, 60:20:15, v/v) and solvent B (methanol: water, 88:12, v/v) as the mobile phase. In all, 0.1% TBME was used for linear gradient elution.

**Hormone content measurement**. Auxins, cytokinins, and gibberellins were measured in extracts of flag leaves sampled at the heading stage during the 2018 field experiment in Beijing. All hormones were measured using Met-Ware (http://www.metware.cn/) based on the AB Sciex Q-TRAP® 4500 LC-MS/MS platform. Three replicates of WT and each transgenic line were performed.

**Metabolite profiling**. Metabolite extraction was conducted according to a previously published protocol[63]. Approximately 50 mg ground and frozen rice leaf sample was mixed with 700 µl 100% methanol and 30 µl stock ribitol (0.2 mg ml$^{-1}$ stock in water) and then shaken for 15 min at 70 °C in a thermomixer (Eppendorf). After centrifugation for 10 min at 14,000 rpm, the supernatant was transferred to a new tube. Four hundred µl chloroform and 800 µl water were added, and then the tube was carefully vortexed for 15 s before centrifugation at 14,000 rpm for 15 min. Two aliquots of 50 µl each were taken from the upper polar phase and transferred to 1.5 ml fresh Eppendorf-tubes before drying in a speed vac (Eppendorf) overnight at room temperature. The fraction enriched in polar primary metabolites was prepared and subsequently processed by routine gas chromatography–mass spectrometry (GC-MS; Agilent) profiling analysis. After derivatization, extracted samples were detected by GC-time-of-flight-MS. Data processing was performed using the TagFinder software. All metabolomic data were divided by the median of each metabolite over all tested samples, and normalized data were used for further analysis. Heatmaps and hierarchical cluster analysis were generated by Multi-Experiment Viewer (version 4.8.1).

**Statistics and reproducibility**. Significance analysis of all experimental data between WT and transgenic lines (including CK = null segregants isolated from selfed heterozygous transgenic plants) were determined according to two-tailed Student's $t$ test using Microsoft Excel. No outliers were excluded in any statistical analysis. One-way analysis of variance test was performed using SigmaPlot v12.5 (SYSTAT, CA, USA). Figures were generated using SigmaPlot v12.5 and GraphPad Prism 7 (GraphPad, CA, USA). Details of the number of biological replicates are described in the figure legends.

**Reporting summary**. Further information on research design is available in the Nature Research Reporting Summary linked to this article.

## Data availability

The authors confirm that the data supporting the findings of this study are available within Supplementary Materials.

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

## Acknowledgements

W.Z. was supported by the Innovation Program of Chinese Academy of Agricultural Sciences, the Elite Youth Program of the Chinese Academy of Agricultural Science, and by grants from the National key research and development program China (2016YFD0300102) and the National transgenic major project of China (2018ZX08010–07B). X.L. was funded by the National Natural Science Foundation (31601237). P.W. and J.A.L were supported by a grant from the Bill & Melinda Gates Foundation (C₄ Rice Phase II) awarded to the International Rice Research Institute. We thank Dr. Ralph Bock (Max Planck Institute of Molecular Plant Physiology) for thoughtful discussions throughout this research.

## Author contributions

Experiments were designed by X.L., P.W., and W.Z. Experiments were performed by X.L., P.W., J.L., S.W., Y.Y., J.Y., M.Z., and W.Z. Data analysis was performed by X.L., P.W., J.A.L., and W.Z. The manuscript was prepared and edited by X.L., P.W., J.A.L., and W.Z. All authors read and approved the final manuscript.

## Competing interests

The authors declare no competing interests.
