## [Peer Review File · Communications Biology]

Editorial Note: *This manuscript has been previously reviewed at another Nature Research journal. This document only contains reviewer comments and rebuttal letters for versions considered at Communications Biology*

REVIEWERS' COMMENTS:

Reviewer #1 (Remarks to the Author):

This is a revision of a manuscript that I have seen before. I have studied the response letter of the authors and had then a fresh look at the revised manuscript without consulting the version with highlighted changes. The reason is that after addition of revisions suggested by different reviewers a paper can become quite distorted.

The new manuscript is easy to read. There are some language issues which can be easily solved by having a native speaker reading it through one more time. No need for professional language editing. Figures and supplementals are fine. The main message is that GLK overexpression enhances yield in rice. This effect is not unexpected giving previous results from other groups. It is also not clear what the molecular cause(s) are for this. The authors identified a number of candidates and it might be a combination of all of them. Taken together, the authors provide a nice manuscript.

There are two major things that should be included. First, one need to be careful about claims whether such approaches are generally applicable. The best examples are the tobacco VPZ lines published in Science in 2016. The same approach seems not to work in Arabidopsis (Nat Plants. 2020 Jan;6(1):9-12. doi: 10.1038/s41477-019-0572-z. Epub 2020 Jan 6.). This is an important information: the same approach might not work in all species, possibly even only in few of them.

What might be the reason for this? One suggestion is that there might be trade-offs associated with overexpressing GLKs or VPZs. In fact, one needs to ask why nature has not invented this overexpression by itself if it apparently does make plants so much better? Where is here the difference between yield and fitness that should have prevented to let GLKs also become overexpressed in nature? This should be also discussed by the authors at the end of their discussion section to give a greater impact. Indeed, there are some repetitions in the discussion section (also with respect to the introduction) and the discussion can be easily shortened in some parts.

Details:

Line 68: the tobacco VPZ lines are only half of the story. You need to consider also the Arabidopsis VPZ lines here.

Line 81: "Hasn't" is not written English

Line 308: rephrase this sentence

Line 318 and following. This repeats parts of the introduction and with respect to VPZ lines needs to be updated.

Line 334: "occur"

Line 334: the entire sentence needs to be rephrased.

Line 389 and following. To discuss % differences in yield of different experiments is maybe not very straightforward. Instead (or: in addition) discuss please possible trade offs of overexpressing GLKs in light with the experiences of the VPZ approach in different species.

Please find below a point-by-point response to reviewers.

The red text in the revised manuscript indicates changes/additions that were made to the text.

=====

REVIEWERS' COMMENTS:

Reviewer #1 (Remarks to the Author):

This is a revision of a manuscript that I have seen before. I have studied the response letter of the authors and had then a fresh look at the revised manuscript without consulting the version with highlighted changes. The reason is that after addition of revisions suggested by different reviewers a paper can become quite distorted.

The new manuscript is easy to read. There are some language issues which can be easily solved by having a native speaker reading it through one more time. No need for professional language editing. Figures and supplementals are fine. The main message is that GLK overexpression enhances yield in rice. This effect is not unexpected giving previous results from other groups. It is also not clear what the molecular cause(s) are for this. The authors identified a number of candidates and it might be a combination of all of them. Taken together, the authors provide a nice manuscript.

Answer: We thank the referee for his/her support for our manuscript.

There are two major things that should be included. First, one need to be careful about claims whether such approaches are generally applicable. The best examples are the tobacco VPZ lines published in Science in 2016. The same approach seems not to work in Arabidopsis (Nat Plants. 2020 Jan;6(1):9-12. doi: 10.1038/s41477-019-0572-z. Epub 2020 Jan 6.). This is an important information: the same approach might not work in all species, possibly even only in few of them. What might be the reason for this? One suggestion is that there might be trade-offs associated with overexpressing GLKs or VPZs. In fact, one needs to ask why nature has not invented this overexpression by itself if it apparently does make plants so

much better? Where is here the difference between yield and fitness that should have prevented to let GLKs also become overexpressed in nature? This should be also discussed by the authors at the end of their discussion section to give a greater impact. Indeed, there are some repetitions in the discussion section (also with respect to the introduction) and the discussion can be easily shortened in some parts.

Answer: We added the newly published results on *Arabidopsis* VPZ lines and discussed the different performance between tobacco and *Arabidopsis* VPZ lines at the end of the discussion section (Line 370-373). We have also removed the repetitive parts of the introduction and discussion (crossed though with a red line).

Details:

Line 68: the tobacco VPZ lines are only half of the story. You need to consider also the *Arabidopsis* VPZ lines here.

Answer: The information on *Arabidopsis* VPZ lines was added (Line 68-73).

Line 81: “Hasn’t” is not written English

Answer: Corrected (Line 84).

Line 308: rephrase this sentence

Answer: We rephrased the sentence (Line 303-307).

Line 318 and following. This repeats parts of the introduction and with respect to VPZ lines needs to be updated.

Answer: We discussed the different performance of tobacco VPZ lines and *Arabidopsis* VPZ lines at the end of the discussion section (Line 370-373).

Line 334: “occur”

Line 334: the entire sentence needs to be rephrased.

Answer: We corrected the spelling mistake and rephrased the sentence (Line 322-324).

Line 389 and following. To discuss % differences in yield of different experiments is maybe not very straightforward. Instead (or: in addition) discuss please possible trade-offs of overexpressing GLKs in light with the experiences of the VPZ approach in different species.

Answer: We rephrased the sentence and added more discussion about the trade-off

of overexpressing GLKs based on the VPZ lines (Line 370-373; Line 358-360; Line 378).

Editorial suggestion:

Please note that titles should be in present tense and ideally should be a declarative sentence. The title should be no more than ~15 words.

We recommend the following title:

< Maize GOLDEN2-LIKE genes enhance biomass and grain yields in rice by affecting photosynthesis and photoinhibition >

Answer: we changed and shortened the title to 16 words (Line 1-3).